# Maternal Renal Dysfunction in Late Pregnancy in Twin and Singleton Pregnancies: Retrospective Study

**DOI:** 10.3390/jcm12010090

**Published:** 2022-12-22

**Authors:** Aki Minoda, Hiroyuki Tsuda, Yoshiki Masahashi, Takuto Nakamura, Miho Suzuki, Nobuhiko Fukuhara, Yumiko Ito, Atsuko Tezuka, Tomoko Ando, Kimio Mizuno

**Affiliations:** Department of Obstetrics and Gynecology, Japanese Red Cross Aichi Medical Center Nagoya Daiichi Hospital, Nagoya 453-8511, Japan

**Keywords:** chorionicity, renal function, serum creatinine concentration, singleton pregnancy, twin pregnancy

## Abstract

This study aimed to evaluate the differences in the impact on maternal renal function between singleton and twin pregnancies in the second half of pregnancy. It retrospectively enrolled 1711 pregnant women consisting of 1547 singleton pregnancies and 164 twin pregnancies from Japanese Red Cross Aichi Medical Center Nagoya Daiichi Hospital from January 2019 to June 2021. Patients underwent renal function tests (serum blood urea nitrogen, creatinine, and estimated glomerular filtration rate (eGFR)) at least one month before delivery. The main outcome measure was maternal renal dysfunction, defined as a serum creatinine level above 0.8 mg/dL. The serum creatinine level was significantly higher and the eGFR was significantly lower in twin than in singleton pregnancies (*p* < 0.001). In addition, the rate of renal dysfunction was significantly higher in twin than in singleton pregnancies (7.9% vs. 2.6%; *p* < 0.01). Multivariate analysis revealed that twin pregnancy (odds ratio (OR) 3.38), nulliparity (OR 2.31), and preeclampsia (OR 3.64) were significant risk factors for maternal renal dysfunction. Maternal renal dysfunction was observed in 13 twin pregnancies, all of which recovered to within normal limits during the early months of the postpartum period. Twin pregnancy is a significant risk factor for maternal renal dysfunction; renal function should be carefully monitored in twin pregnancies.

## 1. Introduction

In pregnancy, both structural and functional changes occur in the maternal kidneys. In terms of structural changes, the length of both kidneys increases by 1 to 1.5 cm and the kidney volume increases by up to 30% [1]. Functional changes manifest as an increase in the renal plasma flow (RPF) and glomerular filtration rate (GFR) [2,3]. The increase in the latter leads to a decrease in serum creatinine concentrations during early pregnancy [4]. A retrospective database study based in Canada demonstrated how the mean serum creatinine concentration decreases in the first trimester of pregnancy, levels off in the second, and then gradually increases in the third trimester to reach near pre-pregnancy levels [5]. Thus, a serum creatinine of 0.8 mg/dL (70.7 µmol/L) or higher, which may be normal in non-pregnant women, is usually indicative of renal impairment in pregnant women, as even slight increases in serum creatinine can reflect a significant decrease in renal function. It is, therefore, necessary to monitor serum creatinine levels during pregnancy to note even mild variations which can help in detecting renal impairment.

The increase in GFR during gestation is primarily due to an increase in RPF [6]. RPF increases by up to 80% at 12 weeks of gestation [7] but decreases to near pre-pregnancy levels in the third trimester. This increase in both parameters is a subsequent result of hemodynamic changes, namely increased cardiac output and decreased systemic vascular resistance, leading to systemic vasodilation [8]. Systemic vasodilation in pregnancy is mainly governed by the renin-angiotensin-aldosterone system. Angiotensin-1 receptors, which are normally involved in vasoconstriction, become downregulated, while angiotensin-2 receptors, which are normally involved in vasodilation, become upregulated. Moreover, there is a noted increase in the levels of angiotensin-2 and aldosterone [9]. Other mediators such as progesterone, nitric oxide, and relaxin have also been reported to play an important role in the hemodynamic changes affecting kidney function during pregnancy [8,9,10,11]. Following its significant rise in the first trimester, RPF begins to decline in the second trimester and then rapidly falls in the third trimester [8,12]. As a result, the filtration fraction and proteinuria increase after 20 weeks of gestation [8,13]. Other factors that can aggravate maternal renal dysfunction during pregnancy involve the mass effect of the enlarged uterus causing compression of the inferior vena cava (IVC) leading to decreased cardiac output, compression of the renal vein, and impaired ureteral transit due to ureteral dilation in the second half of pregnancy.

Considering that maternal renal function is affected by these various physiological and physical factors during pregnancy, a different situation is expected in multiple pregnancies than in singleton pregnancies. To the best of our knowledge, no relevant scientific reports investigating this issue have been published so far. Therefore, in this study, we aimed to investigate any differences regarding maternal renal function between singleton and twin pregnancies in the second half of pregnancy.

## 2. Materials and Methods

### 2.1. Participants

This retrospective study involved 2676 deliveries performed at Japanese Red Cross Aichi Medical Center Nagoya Daiichi Hospital from January 2019 to June 2021. Preoperative tests such as blood tests (complete blood counts, biochemical tests including kidney and liver function tests, coagulation studies, and infectious disease screening including HIV), chest radiography, and electrocardiography were performed for all twin pregnancies and patients with a strong indication for cesarean section delivery within at least one month of planned delivery. After excluding 965 patients (including 959 cases in which blood tests were not performed, 3 cases of missing data, and 3 cases of triplet pregnancy), a final total of 1711 patients were enrolled in the study consisting of 1547 singleton pregnancies and 164 twin pregnancies, all of whom underwent the preoperative blood tests including renal function tests (serum blood urea nitrogen (BUN), creatinine, and estimated glomerular filtration rate (eGFR)) within at least one month of delivery (Figure 1). The study conformed to the principles outlined in the Declaration of Helsinki of 1964 and was approved by the Ethics Committee of the Japanese Red Cross Aichi Medical Center Nagoya Daiichi Hospital, Nagoya, Japan (approval number: 2021-422).

### 2.2. Case Selection and Definition

During each patient visit, the blood pressure was measured, a urinalysis was performed, and ultrasound measurements were taken including the fetal position, fetal growth, and amniotic fluid volume. Screening for congenital anomalies was performed on an as-needed basis. All maternal data including ultrasound findings were stored in medical records. For twin pregnancies, an ultrasound was performed in the first trimester to determine whether the pregnancy was di-chorionic or mono-chorionic twins. That is, an intertwin membrane with the “twin peak” or “lambda” sign indicates dichorionic twins, while that with the “T” sign indicates monochorionic twins [14]. Maternal renal dysfunction was defined as a serum creatinine level above 0.8 mg/dL. Preeclampsia was defined as the occurrence of gestational hypertension (systolic blood pressure >140 mmHg or diastolic blood pressure >90 mmHg after 20 weeks of gestation), proteinuria (>300 mg on 24 h urine collection), and/or signs of end-organ dysfunction during pregnancy [15]. Fetal growth restriction was defined as a birth body weight of <−1.5 SD for gestational age in Japan [16]. In this study, cases of hypertension plus FGR were included in the preeclampsia group and not in the FGR group.

### 2.3. Statistical Analyses

Clinical data extracted from medical records were entered into a computerized spreadsheet (Excel, Microsoft Japan Co., Ltd., Tokyo, Japan). EZR software (version 1.38, Saitama, Japan) was used to perform all data analyses. After using Shapiro–Wilk test to assess the normality of the data, the Mann–Whitney *U* test was conducted to compare continuous variables between the two groups. Student’s *t*-test was used where appropriate. Continuous variables included maternal age, body mass index, birth weight, gestational age at blood test, serum BUN, creatinine, and eGFR. Furthermore, the chi-square test was used to compare the following categorical variables: nulliparity, diabetes mellitus, preeclampsia, fetal growth restriction, and maternal renal dysfunction. A logistic regression model was used to develop the prediction model for maternal renal dysfunction. This included maternal age, body mass index, nulliparity, gestational age at blood test, twin pregnancy, diabetes mellitus, preeclampsia, and fetal growth restriction. The odds ratio (OR) and 95% confidence interval (95% CI) were estimated for each variable. Statistical significance was set at *p* < 0.05.

## 3. Results

### 3.1. Maternal Characteristics and Renal Function between Singleton and Twin Pregnancies

Maternal characteristics are shown in Table 1. The maternal age, pre-pregnancy body mass index, and nulliparity rate were not significantly different between singleton and twin pregnancies. However, the rate of fetal growth restriction was significantly higher in twin pregnancies than that in singleton pregnancies (*p* < 0.001). A comparison of maternal renal function between singleton and twin pregnancies is presented in Table 2. The gestational age at blood test was significantly younger in twin pregnancies than that in singleton pregnancies (*p* < 0.001). Moreover, the serum creatinine levels were significantly higher and eGFR was significantly lower in twin pregnancies than that in singleton pregnancies (*p* < 0.001). In addition, the rate of renal dysfunction was significantly higher in twin pregnancies compared to singleton pregnancies (7.9% vs. 2.6%; *p* < 0.01).

### 3.2. Risk Factors for Maternal Renal Dysfunction Using Multivariate Logistic Regression Analysis

The results of the multivariate analysis of the risk of maternal renal dysfunction are shown in Table 3. The variables included in the analysis were the maternal age, body mass index, nulliparity, gestational age at blood test, twin pregnancy, diabetes mellitus, preeclampsia, and fetal growth restriction. Twin pregnancy (OR 3.38), nulliparity (OR 2.31), and preeclampsia (OR 3.64) were shown to be significant risk factors for maternal renal dysfunction.

### 3.3. Maternal Renal Function between Singleton and Twin Pregnancies Excluding Preeclampsia Cases

Since preeclampsia is an important risk factor that has a significant impact on maternal renal function, we again performed the study after excluding 112 cases of preeclampsia. As a result, the serum creatinine levels were significantly higher (0.59 vs. 0.53; *p* < 0.001) and eGFR was also significantly lower (101.3 vs. 110.8; *p* < 0.001) in twin pregnancies than that in singleton pregnancies. In addition, the rate of renal dysfunction was significantly higher in twin pregnancies compared to singleton pregnancies (7.2% vs. 2.2%; *p* < 0.01).

### 3.4. Renal Function between Dichorionic and Monochorionic Twin Pregnancy

Comparison of renal function between dichorionic and monochorionic twin pregnancies is shown in Table 4. The serum creatinine levels were significantly higher and eGFR was significantly lower in dichorionic twins (*p* < 0.05) compared to monochorionic twins; however, the mean values were within the normal range. However, due to the significantly earlier number of weeks tested in monochorionic twins, both serum creatinine levels and eGFR were no longer significant when analyzed after adjusting for the number of weeks (*p* = 0.117 and *p* = 0.372, respectively). Maternal renal dysfunction was observed in 13 twin pregnancies: 10 (9.1%) dichorionic twins and 3 (5.8%) monochorionic twins. The rate of maternal renal dysfunction was not significantly different between dichorionic and monochorionic twins (*p* = 0.552).

### 3.5. Prognosis in Cases of Maternal Renal Dysfunction in Twin Pregnancy

As described above, maternal renal dysfunction was observed in 13 twin pregnancies in the present study. In 10 of these cases, the renal function normalized within one month postpartum. In the remaining three cases, the renal function also normalized within no more than three months postpartum and required no medical intervention (two recovered within two months postpartum and one case recovered within three months postpartum).

## 4. Discussion

Multivariate analysis showed that twin pregnancy, nulliparity, and preeclampsia were significantly associated with maternal renal dysfunction during pregnancy. While the association with preeclampsia is not surprising, this study highlighted twin pregnancy as a risk factor for maternal renal dysfunction. Conversely, there were no statistically significant differences in maternal renal function between monochorionic and dichorionic twin pregnancies. Herein, all 13 twin pregnancies complicated by renal dysfunction experienced recovery to normal limits within the early months of the postpartum period. These results have important implications in clinical settings as they highlight the need for careful monitoring of worsening renal function in twin pregnancies, especially in cases of preeclampsia.

Pregnancy leads to both structural and functional changes in the maternal kidneys. During the first half of pregnancy, the maternal cardiac output, RPF, and GFR increase while systemic vascular resistance decreases; as a result, the mean serum creatinine concentration is expected to be lower than that of non-pregnant women [4,7,8]. Several mechanisms have been proposed to explain the observed decrease in systemic vascular resistance during pregnancy. One mechanism involves the increase in expression of angiotensin-2 receptors, which normally lead to vasodilation when activated by angiotensin-2 [17]. Another mechanism involves an increase in the synthesis of nitric oxide, a mediator of systemic and renal vasodilation [10]. Moreover, relaxin, a hormone secreted in large amounts by the placenta and decidua in response to human chorionic gonadotropin during pregnancy, increases endothelin and nitric oxide production in the renal circulation, leading to generalized renal vasodilation, decreased renal afferent and efferent arteriolar resistance, and a subsequent increase in renal blood flow and GFR [18]. Conversely, during the second half of pregnancy, RPF decreases slightly, resulting in an increase in filtration fraction and proteinuria [8,12,19,20]. In addition, the physical pressure exerted by the enlarged uterus on blood vessels and organs, particularly during the late pregnancy period, may also negatively impact the maternal renal function. However, there is no consensus among the different studies regarding the increase in filtration fraction seen during the third trimester of pregnancy, and the mechanisms that drive the physiological increase in urinary protein excretion during pregnancy are not well-understood.

To our knowledge, this is the first study showing twin pregnancy to be a significant factor associated with maternal renal dysfunction during the second half of pregnancy. A case series regarding pregnancies after renal transplantation concluded that it is necessary to define more strict criteria for the management of women with twin pregnancies to ensure better maternal outcomes [21,22], which supports our results in this study. In the remainder of this section, we discuss how twin pregnancies can negatively impact maternal renal function. First, the cardiac output of women with twin pregnancies is 20% higher than that in singleton pregnancies and peaks at 30 weeks of gestation [23]. Conversely, in a normal singleton pregnancy, lying in the supine position compared to the left lateral decubitus position can lower cardiac output by as much as 25–30% due to compression of the IVC by the gravid uterus [24]. Therefore, a larger uterus, as seen in a twin pregnancy, can lead to more pronounced IVC compression with a greater resulting reduction in cardiac output and RPF when assuming the supine position for longer periods of time. The reduction in RPF can lead to a decrease in GFR and a subsequent worsening of renal function. However, there were no data on the posture of the pregnant women in this study, and future studies are needed to investigate this aspect.

Second, most pregnant women develop hydroureter and/or hydronephrosis during their pregnancy, which are usually more pronounced on the right side rather than the left [25]. The development of these entities during pregnancy has been attributed to hormonal influences, external compression, and intrinsic changes in the ureteral wall [26]. Progesterone plays a role in reducing ureteral tone, peristalsis, and contraction pressure. Such effects are more prominent in twin pregnancies. Moreover, as pregnancy advances, the enlarged uterus may cause the ureters to become elongated, tortuous, and laterally displaced. As previously mentioned, twin pregnancies are more likely to cause compression on maternal organs due to the mass effect, and this includes the urinary tract system. Occasionally, obstruction of the ureters by the uterus is sufficient to cause kidney failure [27]. For instance, there have been reports of acute kidney injury secondary to urinary tract obstruction caused by enlarged uterine fibroids during pregnancy [28]. In such cases, the insertion of a ureteric stent or delivery of the fetus can relieve the obstruction [29]. In short, twin pregnancies are more likely to cause urinary tract obstruction through hormonal and/or mass effects, and can, therefore, lead to maternal renal dysfunction. Third, during the third trimester, proteinuria increases, and the rate of pathological proteinuria (300 mg/day or greater) is greater in twin pregnancies than that in singleton pregnancies (adjusted OR = 9.13) [20]. In recent years, the urine albumin-to-creatinine ratio (UACR) has been introduced as a useful predictive test for detecting significant proteinuria. The diagnostic accuracy of the UACR (using a threshold between 20 and 60 mg albumin/g creatinine) and the urine protein-to-creatinine ratio (UPCR) (>300 mg protein/day by 24-h urine collection) are approximately equal, and the UACR is rapidly being replaced by UPCR [30,31,32]. Increased albuminuria reflects impaired permselectivity of glomerular capillaries to macromolecules, which is reported to be a marker of kidney damage [33]. Therefore, the pathologic increase in proteinuria (albuminuria) seen in twin pregnancies is reflective of kidney damage and can subsequently lead to maternal renal dysfunction. In this study, even for cases of renal dysfunction, quantitative urine protein testing was not performed if the urine protein qualitative test was negative. Further investigation into the relationship between proteinuria (albuminuria) and renal dysfunction is warranted. Lastly, Cappai et al. [34] reported that the circulation creatinine levels were significantly different between single and twin kidding does in Sarda dairy goats, and electrolyte levels such as sodium, chloride, magnesium, and potassium were also different in the final third of gestation. Differences in the metabolic state and distribution of bodily fluids may explain the varying impact on renal function between singleton and twin pregnancies and should be further studied.

This study has several limitations. In this study, the cutoff value for serum creatinine was set at 0.8 mg/dL based on the upper limit of the normal range of our laboratory data. However, Harel Z et al. [5] and Lopes van Balen VA et al. [35] adopted a lower cutoff value of 0.67 mg/dL and 0.75 mg/dL, respectively. Therefore, the results may differ depending on the cutoff value that defines renal dysfunction. Thus, the results should be further validated based on the reference values for testing at each institution. Next, the selection of the singleton pregnancy group involved only the cases that underwent renal function tests prior to delivery. All twin pregnancies are subjected to blood sampling, but not all cases of singleton pregnancies are sampled. Therefore, selection bias may exist. Moreover, the maternal renal function tests reviewed in this study were supposed to be based on blood sample test results taken closest to the time of delivery. However, most cases were tested only once, and data over the course of the pregnancy were not reviewed. Almost all blood tests for maternal screening were completed prior to the planned cesarean section date. Since twin pregnancies are usually delivered earlier than singleton pregnancies, testing was also performed earlier. In Japan, guidelines (JSOG guidelines) define FGR as an estimated fetal weight of −1.5 SD or less, but these criteria do not distinguish between singleton and multiple pregnancies. We believe that twin pregnancies tend to have smaller fetuses than singleton pregnancies, thus resulting in a significantly higher rate of FGR. Furthermore, cardiac output during pregnancy is reported to vary greatly with the maternal posture (e.g., time spent in the supine position) and uterine size (e.g., amniotic fluid volume); however, we presented no quantitative data on these factors. Lastly, we presented no data regarding proteinuria (albuminuria). In future studies, we would like to examine the albuminuria and cystatin C levels, which are markers of renal insufficiency [31], to have a more global exploration of the factors at play in maternal renal dysfunction during pregnancy.

## 5. Conclusions

In conclusion, twin pregnancy was found to be a significant risk factor for maternal renal dysfunction (defined as serum creatinine > 0.8 mg/dL). The rate of maternal renal dysfunction was not significantly different between dichorionic and monochorionic twins. In 13 twin pregnancies with renal dysfunction, the renal function of all women normalized shortly after delivery, without requiring any medical intervention. This suggests that the negative impact of pregnancy on maternal renal dysfunction is more pronounced in twin pregnancies. Hence, it is important to carefully monitor the renal function for any sign of deterioration in the setting of twin pregnancies. Finally, we hope that further studies will be conducted to better elucidate the physiology of maternal renal function during pregnancy.

## Figures and Tables

**Figure 1 jcm-12-00090-f001:**
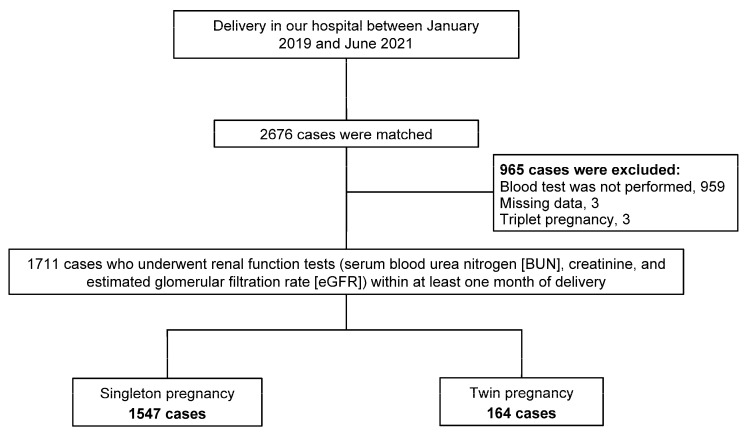
Flowchart of the study.

**Table 1 jcm-12-00090-t001:** Maternal characteristics of this study.

	Singleton Pregnancy (*n* = 1547)	Twin Pregnancy (*n* = 164)	*p*-Value
Maternal age (years) *	33 (16–51)	32 (18–45)	0.296
Pre-pregnant BMI (kg/m^2^) *	20.9 (13.7–47.9)	20.6 (12.2–39.4)	0.187
Nulliparous	797/1547 (51.5%)	99/164 (60.3%)	0.061
Diabetes mellitus	68/1547 (4.4%)	6/164 (3.7%)	0.84
Preeclampsia	101/1547 (6.5%)	11/164 (6.7%)	0.869
Fetal growth restriction	119/1547 (7.7%)	33/164 (20.1%)	<0.001

BMI, body mass index; *: median (range).

**Table 2 jcm-12-00090-t002:** Maternal renal function during pregnancy between singleton and twin pregnancies.

	Singleton Pregnancy (*n* = 1547)	Twin Pregnancy (*n* = 164)	*p*-Value
Serum BUN (mg/dL) *	8 (2–27)	7 (2–32)	0.207
Serum creatinine (mg/dL) *	0.52 (0.28–1.57)	0.56 (0.32–1.61)	<0.001
Renal dysfunction **	41/1547 (2.6%)	13/164 (7.9%)	<0.01
eGFR (mL/min/1.73 m^2^) *	108.2 (29.9–220.4)	100.0 (30.9–156.0)	<0.001
Gestational weeks at blood test *	35.7 (22–42)	34.6 (23–37.4)	<0.001

BUN, blood urea nitrogen; eGFR estimated glomerular filtration rate; *: median (range); **: renal function means serum creatinine was above 0.8 mg/dL.

**Table 3 jcm-12-00090-t003:** Multivariate logistic regression analysis for renal dysfunction (defined as serum creatinine > 0.8 mg/dL).

	Adjusted OR	95% CI	*p*-Value
Twin pregnancy	3.38	1.70–6.75	<0.001
Maternal age	1.04	0.984–1.10	0.159
Nulliparous	2.31	1.22–4.40	<0.05
Gestational days at blood test	1.01	0.995–1.02	0.233
Pre-pregnant BMI	0.959	0.890–1.03	0.262
Diabetes mellitus	1.55	0.452–5.30	0.487
Preeclampsia	3.64	1.75–7.56	<0.001
Fetal growth restriction	0.912	0.367–2.26	0.842

BMI, body mass index; OR, odds ratio; CI, confidence interval.

**Table 4 jcm-12-00090-t004:** Renal function during pregnancy between dichorionic and monochorionic twin pregnancies.

	Dichorionic Twin (*n* = 110)	Monochorionic Twin (*n* = 54)	*p*-Value
Serum BUN (mg/dL) *	8 (2–16)	6 (3–32)	0.27
Serum creatinine (mg/dL) *	0.57 (0.39–1.56)	0.53 (0.32–1.61)	<0.05
Renal dysfunction **	10/110 (9.1%)	3/52 (5.8%)	0.552
eGFR (mL/min/1.73 m^2^) *	98.5 (31.0–156)	106.2 (30.9–153.4)	<0.05
Gestation weeks at blood test *	34.7 (23–37.4)	33.7 (25.4–36.4)	<0.001

BUN, blood urea nitrogen; eGFR estimated glomerular filtration rate; *: median (range); **: renal function means serum creatinine was above 0.8 mg/dL.

## Data Availability

The data (de-identified participant data) that support the findings of this study are available from the corresponding author (HT) upon reasonable request.

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
