# Peer review of "Maternal Renal Dysfunction in Late Pregnancy in Twin and Singleton Pregnancies: Retrospective Study"

_jcm, 2022, doi:10.3390/jcm12010090_

Round 1

Reviewer 1 Report

 Dear Authors,

Many thanks for this piece of work, which I found very interesting and meritorious of being considered further.

Please, trust me when I say that I found some difficulties in following the speech, which despite my expertise in the field, turned out not easy to follow, more than once throughout the text. This is mainly due to the writing style, for which I would suggest you to go through very carefully, because it appears too much chaotic, repetitive and also some sentences fail to correctly display the concepts (they are clear to me, don't want to be misunderstood, but they can be expressed much better).

On the whole, the observations were plausible and also the data provide a large basis for the clinical evaluation  of pregnant women, when the risk factor due to twin pregnancy increases the frequency of failure/insufficient renal clearance.

At this regard, I would also invite the authors to provide a more detailed pathophysiology of the general metabolic milieu of late gestation as to whether facing singleton vs. twins in the discussion session of the paper, because it appears too 'clinical' (which is correct of course) but few 'prodromic' for the final manifestation  of increased creatinine or eGFR I  pregnant patients. I would also highly recommend to refer to the paper by Cappai et al. 2019 Res Vet Sci, 123:84-90, doi: 10.1016/j.rvsc.2018.12.016, where a full description of circulating levels of electrolytes and creatinine values were significantly pointed out in gestating experimental goats carrying out single vs twin pregnancies, for a comparative approach supporting your results.

Thank you.

Author Response

Thank you for reviewing our manuscript and providing us with your valuable comments and useful suggestions, which have helped us improve our paper considerably. As indicated in the responses that follow, we have taken all your comments and suggestions into account in the revision of the manuscript.

Point 1: I found some difficulties in following the speech, which despite my expertise in the field, turned out not easy to follow, more than once throughout the text. This is mainly due to the writing style, for which I would suggest you to go through very carefully, because it appears too much chaotic, repetitive and also some sentences fail to correctly display the concepts.

Response 1: Thank you for this suggestion. In the revised manuscript, we have polished the sentences and ensured that they are logical and easy to read.

Point 2: I would also invite the authors to provide a more detailed pathophysiology of the general metabolic milieu of late gestation as to whether facing singleton vs. twins in the discussion session of the paper. I would also highly recommend to refer to the paper by Cappai et al. 2019 Res Vet Sci, 123:84-90, doi: 10.1016/j.rvsc.2018.12.016, where a full description of circulating levels of electrolytes and creatinine values were significantly pointed out in gestating experimental goats carrying out single vs twin pregnancies, for a comparative approach supporting your results.

Response 2: Thank you for sharing valuable literature that supports our data. It is interesting to see how changes in fluid composition and metabolic status affect renal function in pregnant women. We would like to address this issue in the future. In the revised manuscript, we have introduced this paper and added some comments in the Discussion.

Reviewer 2 Report

The authors compare maternal kidney function among single and twin pregnancies and the incidence of maternal dysfunction. This is an interesting concept and the article is very accessible to the reader. However, corrections are necessary, without which the article cannot be published. Below are my comments Introduction: Ruler 39-40 Why was this creatinine level chosen? According to the authors cited [5], there is even a lower level of creatinine at the moment of measurement chosen by you. Line 57-61 - no citation Lines 62-66 The objectives of the study are double-described and may be difficult to understand. Rephrase please. Methods: Line 69- Hospital XXX? - please specify Figure 1 - You analyzed 2676 deliveries in your hospital. Among the 1711 analyzed, twin pregnancies account for about 10% of all, and even among all selected, they constitute a large group. Why so many twin pregnancies? Line 99-100 - Currently, FGR should be defined according to the 2016 Delphi criteria. The work does not accurately assess this pregnancy complication, so it is not such a big shortcoming. The citation you choose is quite distant in time and you need to consider more recent literature. In my opinion, you should apply the new guidelines. I understand this may be an internal recommendation issue, but please consider this in future research.

Author Response

Thank you for reviewing our manuscript and providing us with your comments and useful suggestions, which have helped us improve our paper considerably. As indicated in the responses that follow, we have taken all your comments and suggestions into account in the revision of the manuscript.

Point 1: Why was this creatinine level chosen? According to the authors cited in [5], for the moment of measurement selected by you, you can speak of a renal dysfunction even at lower creatinine levels.

Response 1: Thank you for your accurate remarks. In this study, the cutoff vale for serum creatinine was set at 0.8 mg/dL. This was because it was based on the upper limit of the normal range of our laboratory data. However, Harel Z et al. and Lopes van Balen VA et al. adopted a lower cutoff value of 0.67 mg/dL and 0.75 mg/dL, respectively. Therefore, results may differ depending on the cutoff value that defines renal dysfunction. Thus, the results should be further validated based on the reference values for testing at each institution. In the revised manuscript, we have added the relevant comments and introduced a paper as reference.

Point 2: The objectives of the study are double-described and may be difficult to understand. Please rephrase.

Response 2: As you indicated, this was a duplicate sentence and has been corrected.

Point 3: Hospital XXX? - please specify.

Response 3: Thank you for pointing out this error. We have listed the name of the facility in the revision.

Point 4: You have analyzed 2,676 deliveries in your hospital. Among the 1,711 analyzed, twin pregnancies account for about 10% of all, and even among all selected, they constitute a large group. How it possible you’ve had so many twin pregnancies?

Response 4: Our hospital is a tertiary perinatal center and handles about 1,500 deliveries per year. The subjects of this study were limited to cases in which maternal renal function tests (i.e., blood tests) were performed within one month of delivery. All twin pregnancies are subjected to blood sampling, but not all cases of singleton pregnancies are sampled.

Therefore, the percentage of twin pregnancies is higher because the data do not reflect all singleton pregnancies during the study period. We have added a relevant comment in the revised manuscript.

Point 5: FGR should be defined according to the 2016 Delphi criteria. In my opinion, you should apply the new guidelines. I understand this may be an internal recommendation issue, but please consider this in future research.

Response 5: Thank you for your useful suggestion. In Japan, guidelines (JSOG guideline) recommend that an estimated fetal body weight of -1.5 SD or less should be managed as FGR, but it is important to evaluate it based on new international criteria such as the Delphi criteria. We hope to use them in future studies.

Point 6: Line 127 - different p value than in the table.

Response 6: Thank you for pointing this out. The P < 0.01 was correct.

Point 7: Do you have the criteria to diagnose FGR in twin pregnancies? It is not mentioned anywhere. However, I understand that it may be the same situation as the situation I described in the methods section.

Response 7: In Japan, guidelines (JSOG guidelines) define FGR as an estimated fetal weight of -1.5 SD or less, but these criteria do not distinguish between singleton and multiple pregnancies. We believe that twin pregnancies tend to have smaller fetuses than singleton pregnancies, thus resulting in a significantly higher rate of FGR. We have added relevant comments in the revised manuscript.

Point 8: Table 4 - You are comparing twin pregnancies, but the timing of blood collection for testing is different. It should be noted that creatinine and GFR levels may vary due to later blood sampling in dichorionic pregnancies, because as you yourself have indicated, creatinine and GFR levels gradually return to baseline values ​​in the third trimester.

Response 8: Thank you for your critical comment. We did not think there was a significant problem with serum creatinine and eGFR since both values were within the normal range, although there was a significant difference. However, as you pointed out, we have corrected for the number of weeks and found no significant difference between the two. I have added the above in the Result section in the revised manuscript.

Point 9: In the results section, it is necessary to add data for renal dysfunction when you exclude preeclampsia-complicated pregnancies from the analysis.

Response 9: Thank you for your useful suggestion. We have compared maternal renal function between single and twin pregnancies after excluding 112 cases of preeclampsia. The results showed that twin pregnancies had significantly higher serum creatinine levels and lower eGFR, even after excluding cases of preeclampsia. These results indicate that twin pregnancy itself is associated with maternal renal dysfunction in the second half of pregnancy. In the revised manuscript, we have added the comments in the Results section.

Point 10: Do you have these data and pre-eclampsia was properly diagnosed, or was it diagnosed only on the basis of hypertension and organ or placental failure (in the form of FGR)? This is very unclear to me and it is necessary to clarify this point.

Response 10: Thank you for your comment. We apologize for any misunderstanding. For preeclampsia in this study, both cases of hypertension + proteinuria and cases of hypertension + organ or placental failure were included. Unfortunately, we do not have data on the proportion of these cases. According to the Japanese guidelines (JSOG guideline), a urinalysis must be performed at the time of antenatal checkup, and if the qualitative urine protein test shows 2+ or more, a quantitative urine protein test is required.

In this study, even for cases of renal dysfunction, if the urine protein qualitative test was negative, the urine protein quantification test was not performed, so data on urine protein quantification for all cases are not available. We have added relevant comments in the revised manuscript.

Reviewer 3 Report

dear authors 

thank you for your submission 

we have a few questions that need your clarification 

1. the introduction 

the paper title is about twins; there is no  line about twins, maternal physiology in twins, and types of tiwin???

what was the rationale for performing this study?

line 64; other published studies discussed the impact of twin pregnancy on renal function particularly in patients with kidney transplants

2. methods 

why you have been restricted by operations cases?

why not tack all?? limiting to CS cases may form selection bias

where your cases matched?? in singleton and twin it does not show in the text.

statistics 

table 1 add units to age and BMI please

how is the FGR is only significant while PE ;a main cause of PE is  not??

table 2; I think that both groups should be matched regarding the Gest. age as the changes in pregnant biochemistry is by the days especially in cases of PE and FGR ?? how can you interprtate the significant difference in the gestational age at sampling time???

table 3 ;it is custom for logestic regression and odds ratio to have at least on group as [Refernce group] your table has no ref.gruop???

moreover what is the number of cases that you had analyzed in that table(3)?

kindly explain the contradiction in your result in table 2 and 3 regarding PE  and FGR??

table 4; would you epalin the rationale behind table 4? what was the benefit of di and monochrionic type to your study aim ??? it was not mentioned in your study goal. moreover both subgroups had different gestational age at sampling time.

Discusion 

it need revision ;discuss the impact of your result in more depth ;mention other studies that have addressed twin and renal function 

Gizzo S, Noventa M, Saccardi C, Paccagnella G, Patrelli TS, Cosmi E, D’Antona D. Twin pregnancy after kidney transplantation: what’s on? A case report and review of literature. The Journal of Maternal-Fetal & Neonatal Medicine. 2014 Nov 1;27(17):1816-9.

Sousa MV, Guida JP, Surita FG, Parpinelli MA, Nascimento ML, Mazzali M. Twin pregnancy after kidney transplantation: case report and systematic review. Brazilian Journal of Nephrology. 2020 Jul 15;43:121-6.

Discuss the implication of these result on our daily practice .

References 

kindly update old references .

you 

Author Response

Thank you for reviewing our manuscript and providing your comments and useful suggestions, which have helped us improve our paper considerably. As indicated in the responses that follow, we have taken all your comments and suggestions into account in the revision of the manuscript.

Point 1: The paper title is about twins; there is no line about twins, maternal physiology in twins, and types of twin???. What was the rationale for performing this study?

Response 1: Thank you for your comment. This study examined the physiological differences and effects between singleton and twin pregnancies, with a focus on maternal renal function.

Therefore, "maternal physiology in twins" is the topic of this study. To better clarify the purpose of this study, we have added relevant comments in the revised manuscript.

Point 2: Line 64; other published studies discussed the impact of twin pregnancy on renal function particularly in patients with kidney transplants.

Response 2: Thank you for your suggestion. We have added the papers you mentioned and discussed them in the revised manuscript.

Point 3: Why you have been restricted by operations cases? Limiting to CS cases may form selection bias. where your cases matched?? in singleton and twin it does not show in the text.

Response 3: The cases registered in this study are those in which maternal renal function tests were performed by blood sampling. Since blood sampling for maternal renal function testing is usually limited to preoperative testing for cesarean section, twin pregnancies were included because blood sampling was performed in all cases, while single pregnancies were only partially included because blood sampling was not performed in all cases. Thus, the possibility of selection bias may exist. We have added a relevant comment in the revised manuscript. Since we did not match the cases in this study, we performed a multivariate logistic analysis.

Point 4: Table 1 add units to age and BMI.

Response 4: We have added in the revised manuscript.

Point 5: How is the FGR is only significant while PE.

Response 5: Thank you for your critical question. In Japan, guidelines (JSOG guidelines) define FGR as an estimated fetal weight of -1.5 SD or less, but these criteria do not distinguish between singleton and multiple pregnancies. We believe that twin pregnancies tend to have smaller fetuses than singleton pregnancies, thus resulting in a significantly higher rate of FGR. We have added relevant comments in the revised manuscript.

Point 6: Table 2; I think that both groups should be matched regarding the Gest. age as the changes in pregnant biochemistry is by the days especially in cases of PE and FGR ?? how can you interpret the significant difference in the gestational age at sampling time???

Response 6: In Table 2, we believe that the higher serum creatinine in the twin group despite the earlier gestational weeks at blood sampling suggests that renal dysfunction appears earlier in twins and therefore more rapid. In Table 3, we performed a multivariate logistic analysis that also considers GA, preeclampsia, and FGR and shows that twin pregnancy is a significant risk factor for renal dysfunction.

Point 7: Table 3 ;it is custom for logestic regression and odds ratio to have at least on group as [Refernce group] your table has no ref.group?? Moreover what is the number of cases that you had analyzed in that table(3)?

Response 7: In Table 3, all nominal variables used were for two groups (yes or no). Therefore, there is no notation of a reference group, since all the odds ratio values listed are for the other control group. Additionally, Table 3 showed the results of the analysis for all cases in this study, so a total of 1711 cases (1547 singletons and 164 twins) were examined.

Point 8: Kindly explain the contradiction in your result in table 2 and 3 regarding PE and FGR?.

Response 8: Thank you for the comment. The results of this study show that although the rate of FGR was higher in the twin group as background (and there was no difference between the two groups for preeclampsia), multivariate logistic analysis revealed that preeclampsia was a significant risk factor for maternal renal dysfunction. In addition, FGR was included in some cases of preeclampsia in this study; cases of hypertension + FGR were included in the preeclampsia group and not in the FGR group. This may have caused a misunderstanding, our apologies. We have added a relevant comment in the revised manuscript.

Point 9: Table 4; would you explain the rationale behind table 4? what was the benefit of di and monochrionic type to your study aim ??? it was not mentioned in your study goal. Moreover both subgroups had different gestational age at sampling time.

Response 9: Thank you for your critical question. The main objective of this study was to determine whether there are differences in maternal renal dysfunction between singleton and twin pregnancies, and not to examine differences by chorionicity. However, since there were a relatively large number of twin cases (164) and there are differences in hormone dynamics between di-chorionic and mono-chorionic fetuses, we considered the possibility that maternal renal dysfunction may be affected by chorionicity. We did not think there was a significant problem with serum creatinine and eGFR since both values were within the normal range, although there was a significant difference. As you pointed out, we have corrected for the number of weeks and found no significant difference between the two. I have added the above in the Result section in the revision.

Point 10: Discussion: it need revision ;discuss the impact of your result in more depth ;mention other studies that have addressed twin and renal function.

Response 10: Thank you for your suggestion. We have added the papers you mentioned and discussed them in the revised manuscript.

Point 11: Discuss the implication of these result on our daily practice.

Response 11: We have added relevant comments in the revised manuscript.

Point 12: kindly update old references.

Response 12: Thank you for your suggestion. We have updated some papers in the revised manuscript.

Round 2

Reviewer 3 Report

dear authors 

thank you for accepting our suggestions 

your manuscript has been well revised